# The Impact of COVID-19 on Physical Activity Behavior and Well-Being of Canadians

**DOI:** 10.3390/ijerph17113899

**Published:** 2020-05-31

**Authors:** Iris A. Lesser, Carl P. Nienhuis

**Affiliations:** Faculty of Health Sciences, Kinesiology Chilliwack campus at CEP, University of the Fraser Valley, 45190 Chilliwack, BC, Canada; carl.nienhuis@ufv.ca

**Keywords:** 2019 novel coronavirus diseases (COVID-19), lifestyle restrictions, physical activity, well-being, anxiety

## Abstract

A global pandemic caused by the novel coronavirus (COVID-19) resulted in restrictions to daily living for Canadians, including social distancing and closure of city and provincial recreation facilities, national parks and playgrounds. The objective of this study was to assess how these preemptive measures impacted physical activity behaviour and well-being of Canadians. An online survey was utilized to measure participant physical activity behavior, nature exposure, well-being and anxiety levels. Results indicate that while 40.5% of inactive individuals became less active, only 22.4% of active individuals became less active. Comparatively, 33% of inactive individuals became more active while 40.3% of active individuals became more active. There were significant differences in well-being outcomes in the inactive population between those who were more active, the same or less active (*p* < 0.001) but this was not seen in the active population. Inactive participants who spent more time engaged in outdoor physical activity had lower anxiety than those who spent less time in outdoor physical activity. Public health measures differentially affected Canadians who were active and inactive and physical activity was strongly associated with well-being outcomes in inactive individuals. This suggests that health promoting measures directed towards inactive individuals may be essential to improving well-being.

## 1. Introduction

In winter 2020, a novel coronavirus (COVID-19) that originated in Wuhan, China, began to present itself in Canada. COVID-19 was classified as a global pandemic by the World Health Organization (WHO) on March 12, 2020 with a decree that preemptive measures be taken to mitigate the viral spread. Canada implemented a number of public health measures such as restricting international travel, closing most non-essential business services (including city and provincial recreation facilities, national parks and playgrounds), self-isolation for those that may have been in contact with COVID-19 and requesting that Canadian citizens stay home as much as possible to delay and mitigate the community-based transmission [1]. In addition to these public health measures, citizens were requested to practice social distancing, which is described as maintaining a distance of 2 m between yourself and other people and avoiding social gatherings, limiting contact with older individuals and those in poor health, avoiding common greetings such as handshakes, and avoiding crowded places and non-essential gatherings [1].

With these public health measures in place it would be expected that Canadians would be faced with changes to their physical activity behavior and associated dimensions of well-being. For instance, the closure of recreation facilities, city parks and playgrounds would require Canadians to be innovative in their physical activity practices. While the public health priority is ascertaining that Canadians stay as safe as possible, the unintended consequences may be a reduction in physical activity and an increase in sedentary behavior, exposing the population to an increased risk and an opportunity for deterioration in chronic health conditions [2]. Despite these potential challenges to physical activity engagement through restricted recreation center access and closure of city parks and playgrounds, as working from home becomes mainstream, opportunities to engage in flexible lifestyles may permit opportunities to more naturally incorporate physical activity into daily living. 

Examining physical activity behavior through a multi-dimensional lens, composed of individual, social and physical environments, such as that portrayed in the social ecological model [3], is necessary in the understanding of determinants that affect physical activity engagement. Individual psychological factors include such factors as confidence and perceived competence [4], while the social environment is inclusive of emotional and logistical support from the home, and lastly the physical environment is inclusive of access to low cost recreational opportunities and outdoor physical activity opportunities [5]. Public health advocates continue to strongly promote physical activity in the home environment to prevent the potential detrimental impact of the COVID-19 protective lifestyle regulations [6], and to ensure that restrictions do not require physical activity be eliminated [7]. 

A noted increase in negative psychological side-effects such as post-traumatic stress syndrome, confusion and anger have been reported as an outcome of the pandemic and associated quarantine [8]. Social isolation, financial uncertainty, job loss and childcare challenges, amongst other factors, may impact various well-being outcomes for Canadians. Physical activity is strongly associated with mental health and wellness. Physically active individuals generally experience less stress, depression and anxiety [9], and physical activity has received attention in recent years as a potential treatment for depression and anxiety in addition to, or in place of, pharmaceuticals [10]. An especially promising benefit of physical activity arises from research done in an outdoor environment with increased nature exposure. Research has indicated that exposure to nature increases positive psychological health outcomes such as happiness, mood and self-esteem, enhances vitality, and reduces stress [11].

The objectives of this study were to gain an understanding of the impact of the global pandemic and public health restrictions on Canadians. Specifically, we aim to report changes since COVID-19 began on physical activity barriers and facilitators and engagement, as well as well-being (anxiety, general mental-health), in active and inactive individuals. Further, we aim to explore differences in outdoor physical activity and nature exposure based on classifications of generalized anxiety and well-being. We hypothesize that COVID-19 would negatively impact physical activity participation overall and that this would be associated with barriers to physical activity. Additionally, we expect this to have a negative impact on Canadian well-being especially amongst those who reduced their physical activity levels. Lastly, we expect that those participants who spend more time being physically active in the outdoors would have greater well-being.

## 2. Materials and Methods

### 2.1. Sample

In order to be a participant in the study, participants had to be over the age of 19 and be a Canadian resident. Participants were recruited through snowball sampling using social media and regular media communications including stories in national and local media. Specifically, the use of personal Twitter, Facebook and LinkedIn accounts of the authors, as well as stories from the University of the Fraser Valley, the local newspaper, a national information resource center, the outdoor recreation council of BC and a national news network were all involved in spreading the word regarding the study and referring participants to the study link. This study received approval from the Human Research Ethics Board at the University of the Fraser Valley and all participants provided online informed consent. While 1366 participants began the online survey, 268 responses were incomplete and therefore removed from the dataset. A resultant participant sample of 1098 Canadian adults were included in this study.

### 2.2. Measures 

Participants completed questionnaires using an online survey software (SurveyMonkey) in April and early May 2020 during the strictest public health restrictions in Canada (nationwide restrictions were in place for 50 days) The survey comprised four parts: demographics, physical activity behaviour, outdoor physical activity, and measures of well-being. Demographic characteristics included age, sex and, marital status. Additionally, occupational status (changes due to COVID-19) and childcare obligations (due to COVID-19) were captured. 

#### 2.2.1. Physical Activity Behaviour

The measure of physical activity was assessed for the week prior and therefore would have been based on physical activity during COVID-19 restrictions. Participants reported their current physical activity levels using the Godin Leisure Questionnaire [12]. In order to assess whether participants were physically active, amounts of reported vigorous and moderate physical activity participation in the Godin questionnaire were used to categorize participants as active or inactive. The cut off for active was >150 min of moderate-vigorous physical activity per week while the related cut off for inactive was <149.9 min of moderate-vigorous physical activity per week based on standard physical activity guidelines for health-related benefits [13]. Further physical activity questions were asked to assess current physical activity behavior. This included: whether physical activity had changed (same, more or less) since restriction onset, the type of physical activity most participated in, location of physical activity (indoors, outdoors or both) and whether the location had changed due to social distancing measures. Barriers and facilitators to physical activity were assessed in reference to the perceived benefits, enjoyment, confidence and planning of physical activity behavior. Additional questions were asked to elucidate the potential impact of social distancing on challenges, support and opportunity for engagement in physical activity.

The Behavioural Regulations in Exercise Questionnaire (BREQ-3) was used to assess participant motivation to exercise [14,15]. Analysis utilized the dichotomy of controlled (external and introjected) and autonomous (intrinsic, identified, and integrated) regulations as commonly done in exercise-based research [16]. Each regulatory form has shown adequate reliability (α > 0.70) and strong test-retest reliability (>0.70) [14,15]. 

#### 2.2.2. Outdoor Physical Activity

Participants reported the number of minutes spent in outdoor physical activity per week along with the importance of nature and whether the activity occurred in a natural environment. Additionally, specific items from the nature relatedness scale (NRS) [11] were utilized based on their impact on physical activity behavior. The NRS asks questions about nature relatedness in the context of measuring an individuals’ affective cognitive and physical relationship with the natural world. Each question is scored on a Likert scale of 1 (disagree strongly) to 5 (strongly agree) and then scored for a measure of nature relatedness. The NRS has been shown to be correlated with behavior, environmental scales and frequency of time in nature [17]. Participants were also requested to report their connectedness to nature on a scale of 1 to 10 using a sliding scale. 

#### 2.2.3. Anxiety

The General Anxiety Disorder-7 (GAD-7) was used to identify participant anxiety [18]. The validity of GAD-7 was substantiated in a primary care sample with a sensitivity value of 0.89 and a specificity value of 0.82 [18]. Löwe et al. [19] also showed supporting evidence that the GAD-7 scale was both reliable and valid. Overall GAD-7 scores as well as classification of anxiety severity was utilized for analysis. 

#### 2.2.4. Well-Being

The Mental Health Continuum (MHC-SF) was used to assess overall well-being as well as emotional, psychological and social well-being [20]. While assessing these three dimensions of well-being, the scale is also used to classify participants as flourishing or languishing. Psychometric analysis has demonstrated strong internal consistency (>0.80) and stable three-month test-retest reliability (>0.68) [21]. An overall MHC score as well as participant classification was utilized for data analysis. 

### 2.3. Statistical Analysis

Demographic characteristics were split by sex and summarized using descriptive statistics. Independent *t*-tests and chi square tests were utilized to compare demographic differences across sex. To analyze physical activity behaviour, associated barriers and facilitators to physical and well-being outcomes, participants were split into inactive and active groups and comparative analysis was conducted utilizing *t*-tests and one-way ANOVA. To further explore differences within the active and inactive groups and associated barriers, facilitators and well-being outcomes, participants were further categorized based on self-reported changes in physical activity behavior. One-way ANOVA and Tukey post-hoc tests were thus conducted to explore differences within participants who became more active, remained the same, or became less active due to COVID-19 restrictions. Lastly, independent sample *t*-tests and one-way ANOVAs were conducted to assess differences between active and inactive participants and associated outdoor physical activity and well-being measures. All statistical analyses were computed utilizing SPSS-25.0 software (SPSS Inc., Chicago, IL, USA) and significance was set at *p* < 0.05. 

## 3. Results

Table 1 lists participant demographics split on the basis of sex. Of the participants, the majority were female (79.3%), married or in a domestic relationship (68.5%), and had completed some or all post-secondary education (64.1%). While most participants reported that they were working full-time (58.3%), the majority of participants indicated that they had experienced a change in their employment status on account of COVID-19 (56.8%). The majority of respondents were currently living in British Columbia (59.7%), and most participants lived in either urban (45.7%) or suburban (40.7%) locations. Finally, 25.3% of the participants indicated increased childcare responsibility due to COVID-19 school and childcare closures. 

### 3.1. Changes to Physical Activity Engagement and Barriers and Facilitators to Physical Activity 

Table 2 summarizes the physical activity behaviours and characteristics between the active and inactive participants. There was a significant (*p* < 0.001) difference between active and inactive participants and physical activity behaviour change since COVID-19 restrictions were put in place. Results showed that 40.5% of inactive individuals became less active while only 22.4% of active individuals became less active. Comparatively, 33% of inactive individuals became more active while 40.3% of active individuals became more active. The majority of the participants in the inactive group participated in walking (57.2%), while most commonly reported activities for the active group included running (32.8%), walking (19.7%) and cycling (14.9%). Since COVID-19 restrictions were in place, 28.3% of inactive participants altered their physical activity type, while 39.6% of active participants maintained their typical physical activity choice.

The majority of participants participated in physical activity either in the home environment or in their neighborhood. Table 3 further summarizes results relating to outdoor physical activity behaviour. While the majority of active and inactive participants participated in physical activity in both indoor and outdoor environments, 41.1% of the inactive population compared to 22.6% of the active population participated in physical activity in the outdoor environment. However, the active population spent significantly more minutes per week engaging in outdoor physical activity (M_diff_ = 97.23, SD = 17.49, *p* < 0.001) and reported significantly greater connectedness to nature (M_diff_ = 3.58, SD = 1.42, *p* = 0.012) and nature relatedness (M_diff_ = 0.17, SD = 0.04, *p* < 0.001) than the inactive population.

Regardless of physical activity level, there were significant differences between those who became more active, stayed the same or were less active since COVID-19 restrictions. Within these differences there were differing roles of physical activity barriers and facilitators (see Table 3 for summary). To explore between group differences, a series of Tukey post hoc tests were conducted after a significant between group difference was found. Results indicated that for both the active and inactive population, there was a significant difference between those that were less active and those that reported the same or more physical activity since COVID-19 restrictions. That is, those that were less active for both inactive and active populations reported significantly less benefit, less enjoyment, less confidence, less support and fewer opportunities to be active. Further, those that were less active indicated significantly more challenges in physical activity engagement. 

In addition to these barriers and facilitators of physical activity, independent sample *t*-tests were conducted to assess differences in motivation during COVID-19 restrictions between active and inactive individuals. Results indicated that the active population reported significantly more autonomous motivation than the inactive population (*p* < 0.001). Comparatively the inactive population indicated significantly more amotivation and external regulation than the active population (*p* < 0.001).

### 3.2. Well-Being and Anxiety Scores Based on Changes to Physical Activity 

An independent sample *t*-test revealed that inactive (M = 47.64, SD = 12.70) participants scored significantly lower (*p* = 0.031) on the mental health continuum than active participants (M = 49.37, SD = 11.77), though was a non-significant difference in generalized anxiety (*p* = 0.455). Further analyses explored well-being differences and changes in physical activity within the active and inactive groups. As shown by Table 4, there were significant differences in well-being outcomes in the inactive population between those who were more active, the same or less active (*p* < 0.001) but this was not seen in the active population. In the inactive population, Tukey’s post hoc test found significant differences in the mental health continuum scores, including social, emotional and psychological domains, as well as GAD-7 scores between those who reported becoming more active or remained the same versus those that were less active. Specifically, the inactive participants that were more active or maintained their activity levels indicated higher levels of social, emotional and psychological health and lower levels of generalized anxiety. 

Analysis indicated significant differences within both active and inactive groups between anxiety levels and number of minutes spent in moderate to vigorous exercise (Inactive: F(3.677) = 3.08, *p* = 0.027; Active: F(3.388) = 4.77, *p* = 0.003). A series of Tukey post hoc tests revealed that inactive participants with mild anxiety were more physically active than participants with moderate anxiety (M_diff_ = 10.06, SE = 3.86, *p* = 0.046), and that active participants with low anxiety were significantly less physically active than participants with severe anxiety (M_diff_ = 101.62, SE = 33.44, *p* = 0.013). Analysis further indicated significant differences, within both active and inactive groups, in the mental health continuum score and number of minutes spent in moderate to vigorous exercise for the inactive participants (F(2.693) = 4.66, *p* = 0.010). Specifically, those classified as flourishing were engaged in more (M_diff_ = 25.78, SE = 9.11, *p* = 0.013) moderate to vigorous exercise than those classified as languishing. 

### 3.3. Well-Being Scores Based on Social Support and Levels of Physical Activity 

Additional analyses were conducted to explore relationships between physical activity, social well-being and the presence or absence of being active with others. There was a significant difference in the amount of physical activity done with others between those who did more, the same or less activity in the inactive population (F(2, 693) = 5.98, *p* = 0.003), but this was not found in the active population (F(2, 398) = 1.25, *p* = 0.287) (Table 5). Tukey post hoc tests showed that those in the inactive population who became more physically active reported more of their physical activity done with others than those who remained the same or were less active. Further tests revealed significant differences in mental health continuum scores based on whether participants participated in physical activity with others or participated alone (Inactive: F(2.646) = 3.97, *p* = 0.019; Active: F(2.373) = 6.05, *p* = 0.003). Specifically, for both the active and inactive populations, those who engaged in physical activity to a moderate amount with others scored higher on the mental health continuum than those who did physical activity alone but this was not shown in those that participated with others to a greater or lesser extent.

### 3.4. Differences in Outdoor Physical Activity and Nature Exposure Based on Classifications of Anxiety and Well-Being 

An additional analysis explored differences between outdoor physical activity behaviour based on levels of anxiety and well-being. Analyses of variance indicated significant differences within the inactive group between classification of anxiety levels (GAD-7) and number of minutes spent in outdoor physical activity (F(3.674) = 4.42, *p* = 0.004). A Tukey post hoc test demonstrated that inactive participants with severe anxiety spent significantly fewer minutes in outdoor physical activity than participants with low (M_diff_ = 88.27, SE = 31.61, *p* = 0.027) and mild anxiety (M_diff_ = 65.24, SE = 23,38, *p* = 0.028). 

Additional tests were conducted to explore the relationships between overall well-being and outdoor physical activity. Analyses of variance indicated significant differences in number of outdoor physical activity minutes per week on the mental health continuum in inactive (F(2.690) = 8.14, *p* < 0.001) and active (F(2.396) = 3.20, *p* = 0.042) participants. For active and inactive participants, those classified as flourishing were significantly more active outdoors than those who scored lower on the mental health continuum. There were also significant differences in nature relatedness and mental health in inactive (F(2.693) = 5.79, *p* = 0.002) and active (F(2.399) = 3.74, *p* = 0.025) participants. For active and inactive participants, those classified as flourishing indicated greater nature relatedness than those who scored lower on the mental health continuum. 

## 4. Discussion

To our knowledge this is the first study to assess the impact of the COVID-19 pandemic and associated public health restrictions on physical activity behavior and well-being in Canadians. Our results illustrate that there are significant differences between inactive and active participants with a greater portion of inactive participants reporting less physical activity and a greater portion of active participants reporting more physical activity since COVID-19. Participants who were more physically active had greater mental health scores, while inactive participants who became more active or did more of their physical activity in the outdoors had lower levels of anxiety.

The multiple dynamic factors that influence physical activity behavior would suggest that the public health restrictions would potentially influence physical activity behavior at multiple levels. Our results illustrate that there are differences between inactive and active participants with 40.5% of inactive participants reportedly engaged in less physical activity and 40.3% of active participants reporting more physical activity since COVID-19. It has been shown in prospective studies that some active individuals increase physical activity engagement in stressful times in an effort to cope, while stress has a negative effect on physical activity among less active individuals during acute stress [22]. Additionally, motivation may play a role in the reduced physical activity engagement among inactive participants [23], as active participants reported significantly more autonomous motivation than inactive participants, while the inactive participants indicated significantly more amotivation than the active participants.

Those that became less active for both inactive and active participants reported significantly less benefit, less enjoyment, less confidence, less support and fewer opportunities to be active. Further, those that were less active indicated significantly more difficulty and challenge in engaging in physical activity since COVID-19. This speaks to the importance of exercise self-efficacy in physical activity behavior change and maintenance. Research indicates that higher levels of self-efficacy for physical activity are associated with a higher likelihood of achieving physical activity recommendations, while having stronger intentions to be active and plan for physical activity engagement [24]. It has also been shown that self-efficacy is fundamental to long term physical activity engagement [25], and therefore those individuals who were less active since COVID-19 restrictions may have lacked the self-efficacy to engage in physical activity in a different environment.

Inactive participants who engaged in more or the same amount of physical activity since COVID-19 restrictions were significantly more likely to do their physical activity with others than active participants. This suggests that social support may be less important for those who are active [26]. Social support may be an important contributor to behavior change in inactive individuals due to the challenges experienced with exercise self-efficacy and motivation early in the behavioral change process [27].

We found, in both active and inactive populations, that those classified with higher anxiety levels indicated less moderate to vigorous exercise than those classified with lower anxiety levels. We further found that inactive participants who became more active or maintained their physical activity levels during COVID-19 had lower anxiety than those who decreased their physical activity levels. As the mental duress reported in this study population is likely entirely or at least partially acute in nature due to COVID-19 [8], physical activity may play a protective role in suppressing the stress response elicited by an overactive sympathetic nervous system response [28]. Specifically, the inactive participants that were more active or maintained their activity levels indicated higher levels of social, emotional and psychological health and lower levels of generalized anxiety. Adults who are regularly physically active experience fewer symptoms of anxiety and depression than their peers [29,30] likely due to changes in biological and psychological mechanisms [31]. Physical activity interventions have largely found a reduction in anxiety among healthy adults [32]. 

Inactive participants with severe anxiety were less physically active outdoors than inactive participants with low or mild anxiety. Further, for active and inactive participants, those classified as flourishing were significantly more active outdoors than those who scored lower on the mental health continuum. Exposure to nature has been associated with improvements in mental health with perceived stress acting as a mediator [33]. When comparing indoor to outdoor physical activity, participants who exercise outdoors reported decreased feelings of tension, confusion, anger and depression while reporting a greater intent to repeat activity when done in the outdoors [34]. Some of the theories behind the impact of nature exposure on stress include attentional restoration and improved cognitive function [35], increased social contact [33] and the fact that more people are physically active when outdoors [36]. A reduction in both state and trait anxiety has been seen in individuals who feel a stronger connection to nature through physical exposure and familiarity with nature [37]. Our results showed that for active and inactive participants, those classified as flourishing indicated greater nature relatedness than those who scored lower on the mental health continuum. Additionally, active participants reported greater connectedness to nature and nature relatedness than the inactive population. Therefore, how much one is connected or appreciates the natural environment interrelates with their well-being. 

The inability to assess pre-COVID-19 anxiety and well-being states limited the capacity for this study to explore changes in well-being due to the onset of public health measures. Further, pre-COVID-19 measures of physical activity behaviour would complement the subjective assessment on changes to physical activity levels. Recruitment methods that utilized a web-based voluntary approach would result in some selection bias. Finally, the varied seasonal transitions from winter to spring across Canadian provinces, as well as the varied public health responses from provincial health offices would impact collective opportunities for outdoor, social and economic activity. 

## 5. Conclusions

Public health measures differentially affected Canadians who were active and inactive and physical activity was strongly associated with well-being in inactive individuals. This suggests that health-promoting measures directed towards increasing physical activity levels in inactive individuals may be essential to improving well-being or alternatively improving well-being of Canadians may be necessary in order to increase physical activity levels. Overall physical activity and specifically outdoor physical activity appears to offer protective benefits in well-being and therefore opportunities to be physically active in the outdoors during public health restrictions should be offered where feasible. 

## Figures and Tables

**Table 1 ijerph-17-03899-t001:** Participant demographics split by sex.

Participant Characteristics	Male *N* (%)	Female *N* (%)	Total *N* (%)	*p*-Value
	215 (19.6)	871 (79.3)	1098 (100)	
Age (mean, SD)	45 ± 16	41 ± 15	42 ± 15	<0.001
Relationship Status				
Married/Domestic	168 (78.1)	574 (65.9)	752 (68.5)	0.015
Widowed/Divorced/Separated	8 (3.7)	79 (9.1)	88 (8.0)	
Single	39 (18.1)	217 (24.9)	257 (23.4)	
Education				
high school	7 (3.3)	29 (3.4)	37 (3.4)	0.040
some post-secondary	44 (20.5)	154 (17.7)	198 (19.0)	
completed post-secondary	101 (470)	391 (44.9)	495 (45.1)	
some graduate	20 (9.3)	56 (6.4)	76 (6.9)	
completed graduate	43 (20.0)	241 (27.7)	292 (26.6)	
Employment status (pre COVID)				
Full-time	144 (67)	485 (55.7)	640 (58.3)	< 0.001
Part-time	13 (6)	189 (21.7)	203 (18.5)	
Unemployed	10 (4.7)	41 (4.7)	51 (4.6)	
Homemaker	1 (0.5)	42 (4.8)	43 (3.9)	
Retired	46 (21.4)	99 (1.4)	145 (13.2)	
Unable to work	0 (0)	10 (1.1)	10 (0.9)	
Employment status (post COVID)				
no change	122 (56.7)	350 (40.2)	473 (43.2)	<0.001
reduced hours	17 (7.9)	93 (10.7)	110 (10)	
remote work	58 (270)	286 (32.8)	352 (32.1)	
laid off	17 (7.9)	141 (16.2)	161 (14.7)	
Environment				
Urban	109 (50.7)	384 (44.1)	502 (45.7)	0.021
Rural	23 (10.7)	121 (13.9)	146 (13.3)	
Suburban	83 (38.1)	364 (41.8)	447 (40.7)	
Location				
West Coast (BC)	129 (60.3)	517 (59.4)	655 (59.7)	0.584
Prairies (AB, SASK, MB)	30 (14)	113 (13.1)	134 (13.1)	
Central (ON, QC)	43 (22)	156 (17.9)	202 (18.4)	
Maritimes (NB, NS, PEI, NFLD)	11 (4.2)	80 (9.1)	92 (8.3)	
Territories	1	2 (0.2)	3 (0.3)	
Childcare				
Yes	51 (23.7)	224 (25.7)	278 (25.3)	0.502
No	164 (76.3)	646 (74.2)	819 (74.7)	

The *p*-values represent chi-square tests of independence indicating associations between sex and categorical variables.

**Table 2 ijerph-17-03899-t002:** Physical activity behaviour and characteristics.

Participant Characteristics	Inactive *N* (%)	Active *N* (%)	*p*-Value
	696 (63.4)	402 (36.6)	
Godin Leisure Score (mean, SD)	92.99 ± 69.46	253.67 ± 166.91	<0.001
Moderate-vigorous Physical Activity (min/week) (mean, SD)	60.71 ± 43.35	301.49 ± 186.09	<0.001
Change in physical activity since COVID-19			
more	230 (33)	162 (40.3)	<0.001
about the same	184 (26.4)	150 (37.3)	
less	282 (40.5)	90 (22.4)	
Most common type of physical activity			
walking	398 (57.2)	79 (19.7)	<0.001
jogging/running	53 (7.6)	132 (32.8)	
biking/cycling	37 (5.3)	60 (14.9)	
weight training	9 (1.3)	17 (4.2)	
online videos/classes	36 (5.2)	44 (10.9)	
yoga	47 (6.8)	5 (1.2)	
home workout	63 (9.1)	43 (10.7)	
hiking	12 (1.7)	6 (1.5)	
home/yard work	13 (1.9)	4 (1.0)	
other	12 (1.7)	11 (2.7)	
Change in type of activity since COVID-19			
very similar	238 (34.2)	159 (39.6)	0.004
somewhat similar	258 (37.1)	163 (40.5)	
not so similar	197 (28.3)	79 (19.7)	
Most common physical activity location			
in the house	218 (31.3)	155 (38.6)	<0.001
neighborhood	259 (37.2)	95 (23.6)	
rec sites, trails, parks	85 (12.2)	61 (15.2)	
outside (unspecified)	85 (12.2)	64 (15.9)	
other	31 (4.5)	21 (5.2)	

The *p*-values represent chi-square tests of independence indicating associations between sex and categorical variables.

**Table 3 ijerph-17-03899-t003:** Outdoor physical activity behaviour.

Participant Characteristics	Inactive *N* (%)	Active *N* (%)	*p*-Value
	696 (63.4)	402 (36.6)	
Physical Activity Location			
Indoors	57 (8.2)	21 (5.2)	<0.001
Outdoors	286 (41.1)	91 (22.6)	
Both	350 (50.3)	289 (71.9)	
Natural environment			
Yes	526 (75.6)	332 (82.6)	0.053
No	136 (19.5)	62 (15.4)	
How essential is nature?			
Very important	420 (60)	263 (65.4)	0.359
Somewhat important	189 (27.2)	94 (23.4)	
Not important	81 (11.7)	44 (10.9)	

The p-values represent chi-square tests of independence indicating associations between sex and categorical variables.

**Table 4 ijerph-17-03899-t004:** Barriers and facilitators to Physical Activity behaviour during COVID-19 based on changes in physical activity.

Change in Physical Activity Since COVID-19	Inactive (*n* = 696)			*p-*Value	Active (*n* = 402)			*p*-Value
	More Active (*n* = 217) M ± SD	Same Active (*n* = 167) M ± SD	Less Active (*n* = 265) M ± SD		More Active (*n* = 152) M ± SD	Same Active (*n* = 142) M ± SD	Less Active (*n* = 83) M ± SD	
Barriers								
Have you found it more challenging to engage in physical activity since COVID-19?	1.85 ^a^ ± 1.12	2.23 ^a,b^ ± 1.22	3.96 ^a,b^ ± 1.19	<0.001	1.93 ^a^ ± 1.26	2.24 ^b^ ± 1.05	3.51 ^a,b^ ± 1.29	<0.001
How difficult is it for you to do physical activity?	2.17 ^a^ ± 1.05	2.37 ^b^ ± 1.13	3.41 ^a,b^ ± 1.22	<0.001	1.89 ^a^ ± 1.08	1.98 ^b^ ± 0.94	3.00 ^a,b^ ± 1.11	<0.001
Facilitators								
How confident are you in doing physical activity?	3.91 ^a^ ± 0.92	3.80 ^b^ ± 1.05	2.63 ^a,b^ ± 1.17	<0.001	4.33 ^a^ ± 0.81	4.28 ^b^ ± 0.84	3.72 ^a,b^ ± 0.98	<0.001
How beneficial is physical activity?	4.42 ^a^ ± 0.69	4.35 ^b^ ± 0.80	4.11 ^a,b^ ± 1.00	<0.001	4.86 ^a^ ± 0.37	4.83 ^b^ ± 0.46	4.61 ^a,b^ ± 0.68	<0.001
How detailed of a plan do you have for physical activity?	2.66 ^a^ ± 1.19	2.59 ^b^ ± 1.23	2.06 ^a,b^ ± 1.09	<0.001	3.49 ^a^ ± 1.12	3.75 ^b^ ± 1.01	3.03 ^a,b^ ± 1.34	<0.001
How enjoyable is physical activity right now?	3.94 ^a^ ± 0.91	3.81 ^b^ ± 0.99	2.84 ^a,b^ ± 1.23	<0.001	4.34 ^a^ ± 0.85	4.33 ^b^ ± 0.78	3.73 ^a,b^ ± 1.08	<0.001
How much support do you have for physical activity right now?	3.53 ^a^ ± 1.12	3.20 ^a,b^ ± 1.24	2.52 ^a,b^ ± 1.10	<0.001	3.71 ^a^ ± 1.12	3.68 ^b^ ± 1.11	3.16 ^a,b^ ± 1.17	<0.001
How many opportunities do you have for physical activity?	3.81 ^a^ ± 0.92	3.65 ^b^ ± 0.96	3.00 ^a,b^ ± 1.02	<0.001	4.14 ^a^ ± 0.88	3.99 ^b^ ± 0.85	3.50 ^a,b^ ± 0.99	<0.001

Scores based on means of scale ranging from 1 (Not at all) to 5 (Extremely). Means in a row sharing superscript ‘a’ or ‘b’ are significantly different from each other.

**Table 5 ijerph-17-03899-t005:** Well-being outcomes in relation to changes in physical activity since COVID-19.

Change in Physical Activity Since COVID-19	Inactive (*n* = 696)			*p-*Value	Active (*n* = 402)			*p*-Value
	More Active (*n* = 217) M ± SD	Same Active (*n* = 167) M ± SD	Less Active (*n* = 265) M ± SD		More Active (*n* = 152) M ± SD	Same Active (*n* = 142) M ± SD	Less Active (*n* = 83) M ± SD	
Mental Health Continuum Score	49.34 ^a^ ± 12.30	50.54 ^b^ ± 11.51	44.42 ^a,b^ ± 13.07	<0.001	48.92 ± 12.50	50.56 ± 10.53	48.13 ± 12.35	0.273
Social	15.83 ^a^ ± 5.59	15.75 ^b^ ± 5.38	13.50 ^a,b^ ± 5.89	<0.001	14.96 ± 5.88	15.63 ± 5.06	14.93 ± 5.36	0.498
Emotional	11.66 ^a^ ± 2.61	12.23 ^b^ ± 2.47	10.62 ^a,b^ ± 3.05	<0.001	11.88 ± 2.60	11.97 ± 2.43	11.47 ± 2.82	0.357
Psychological	21.88 ^a^ ± 5.58	22.51 ^b^ ± 5.59	20.22 _a_^,b^ ± 6.02	<0.001	22.05 ± 5.70	22.97 ± 4.70	21.39 ± 5.97	0.085
GAD -7	9.87 ^a^ ± 4.26	8.83 ^b^ ± 4.71	11.24 ^a,b^ ± 4.66	<0.001	9.65 ± 4.74	9.62 ± 4.76	10.98 ± 4.46	0.063

Means in a row sharing superscript ‘a’ or ‘b’ are significantly different from each other.

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
