# Peer review of "The Impact of COVID-19 on Physical Activity Behavior and Well-Being of Canadians"

_ijerph, 2020, doi:10.3390/ijerph17113899_

Round 1

Reviewer 1 Report

The paper shows how public health measures due to Covid-19 differentially appear to have “affected” Canadians who were active and inactive. In addition, the authors found that physical activity was strongly associated with mental health outcomes in inactive individuals. They conclude by suggesting that health promoting measures directed towards inactive individuals may be essential to improving mental well-being.

Overall, the article is very interesting and report relevant descriptive evidence, which is worth knowing as a possible consequence of Covid-19, especially in developed countries like Canada. However, I do have some suggestions for the authors to make the article more consistent and clearer.

Major comments:

  1. Objectives

In multiple parts of the text, the authors describe that the objectives of the paper are: “to assess how these preemptive measures impacted physical activity behaviour and mental well-being of Canadians”.  In practice, they to do not assess “the impact” (row 70) of the measures since they do not estimate any causal effect. Instead, my understanding is that they describe the following:

  • Self-reported changes after Covid-19 of physical activity by active and inactive people (ex-ante)
  • Self-reported changes after Covid-19 of well-being measures by active and inactive people (ex-ante)
  • Barriers and facilitators to Physical Activity behaviour during COVID-19 based on changes in physical activity.
  • Correlations/relationship between physical activity and well-being, and then more specifically, outdoor physical activities and mental health.

First, I would suggest the authors to try to more consistent and precise about what they do. Since they use interchangeably different terms (like well-being, metal well-being, mental health) but then use the same measurable outcomes, and they have a lot of different variables, I would suggest more clarity in the use of the terms for the factors/outcomes looked at, as well as in the use of their sub-sections’ titles (see below).

Second, row 70-73 report that they also look at barriers and facilitators of the physical activity, but they never describe in details what can be/are those, and which ones they look in their results (Table 3). I also recommend to distinguish between barriers (panel A) and facilitators (panel B) in Table 3.

Third, there is a complete lack of description of the hypothesis the authors have in mind when they look at the data. What we should expect and why (should more/less active people increase/decrease physical activity after Covid-19)? The authors cite different studies, but a more structured discussion on expected hypothesis might help the reader. If there is an hypothesis worthy to test with descriptive data, please describe it in the introduction. If not, then clarify the paper is a description of the data with ex-ante no expectation on what we should expect.

Overall, I would like to see a revised version of the introduction:

  • To describe exactly what the objectives are
  • To explain concisely which variables the authors are exploring: what do the authors use for well-being (only mental well-being?), etc.
  • Hypothesis testing based on the literature cited

  1. Organization of the article

Sample

I would like to see many more details on the sample selection and sample in the Sample sub-section. I would move row 144-156  under Sample. It is OK to leave Table 1 as a result since all the paper is descriptive, but at least how many people are in the sample should be described in the sample sub-section. In addition, the statement “…recruitment through snowball sampling using social media and regular media communications including stories in national and local media” is too general: What was the procedure in practice? How many people were selected, how many reached? How many were excluded because of the criteria? What was the targeted sample? Which social media exactly (internet?)? How do you use stories in the media? …

Related to this, could you explain what does “268 incomplete responses” mean? Do you have 0 refusals? How much incompleteness is there? Did participants receive incentives?

Sub-sections

The titles of the sub-sections 3 could better define what the authors look at, following their new objectives above. For example, Titles 3.3 (you have two titles 3.3, so adjust to 3.4) could be changed to reflect “correlations”.

Results

The most interesting part of “3.3. Outdoor Physical Activity Behaviour and Well-Being” are rows 235-250: this is what describe “correlations”. Why is Table 5 a descriptive table rather than showing those correlations? I would move Table 5 and rows 226-234 in the first part of the results where sample and changes are described, and dedicate the second part of the paper to correlations. Show the correlations results in an additional table.

3. Interpretation

What is really lacking is also more details on how you asked the question to define active/inactive people. Is it based on what they were doing before Covid-19? Was it: in a normal week? In the same week last year? etc. The reference time period is fundamental here to understand how the two groups compare.

In addition is 150 minutes weeekly a normal measure used to define active people? 150 minutes seem pretty high and it seems you are really defining “sporty” people rather than just active. So, I would explain better your choice around this definition, to make sure the reader know how to interpret results.

Minor comments:

  1. Be consistent: maybe always report “recreation facilities, city parks and playgrounds” not one or the others (row 41, 31-32, 9-10)
  2. Move reference [3] after the social ecological model, not at the end of the sentence
  3. April and early May 2020 during the strictest public health restrictions in Canada (row 83) – Please define number of days and when/which restrictions were imposed at the same time
  4. How long was the questionnaire? It seems you asked a lot of questions, maybe describe how long was your survey tool. A strength of the study is that you used well-validated measures. I also like the explanation for each section of measures on how the measures have been used/validated.
  5. Typo: pos hoc tests: post-hoc? Row 92
  6. Typo: row 156 3.1. Physical Activity Engagement and Barriers and Faciltators to Physical Activity
  7. Typo: 3.2. Mental Well-Being and Phyical Activity
  8. Typo: 3.3. Phyical Activity, Social Support and Social Well-Being
  9.  Row 220 repetition …with others to a moderate amount with others scored higher on the mental health continuum
  10. Funding: This research received no external funding. How is this possible if the authors collected data?

Author Response

Reviewer 1 Responses

Comments and Suggestions for Authors

The paper shows how public health measures due to Covid-19 differentially appear to have “affected” Canadians who were active and inactive. In addition, the authors found that physical activity was strongly associated with mental health outcomes in inactive individuals. They conclude by suggesting that health promoting measures directed towards inactive individuals may be essential to improving mental well-being.

Overall, the article is very interesting and report relevant descriptive evidence, which is worth knowing as a possible consequence of Covid-19, especially in developed countries like Canada. However, I do have some suggestions for the authors to make the article more consistent and clearer. 

Major comments:

  1. Objectives

In multiple parts of the text, the authors describe that the objectives of the paper are: “to assess how these preemptive measures impacted physical activity behaviour and mental well-being of Canadians”.  In practice, they to do not assess “the impact” (row 70) of the measures since they do not estimate any causal effect. Instead, my understanding is that they describe the following:

  • Self-reported changes after Covid-19 of physical activity by active and inactive people (ex-ante)
  • Self-reported changes after Covid-19 of well-being measures by active and inactive people (ex-ante)
  • Barriers and facilitators to Physical Activity behaviour during COVID-19 based on changes in physical activity.
  • Correlations/relationship between physical activity and well-being, and then more specifically, outdoor physical activities and mental health.

Thank you for these comments. We agree that this would greatly clarify the objectives of the study. We have re-written the objectives stating at line 71 and it now reads as the following:

“The objectives of this study were to gain an understanding of the impact of the global pandemic and public health restrictions on Canadians. Specifically, we aim to report changes since COVID-19 on physical activity barriers and facilitators and engagement as well as well-being (anxiety, general mental health) in active and inactive individuals. Further, we aim to explore differences in outdoor physical activity and nature exposure based on classifications of generalized anxiety and well-being.

First, I would suggest the authors to try to more consistent and precise about what they do. Since they use interchangeably different terms (like well-being, metal well-being, mental health) but then use the same measurable outcomes, and they have a lot of different variables, I would suggest more clarity in the use of the terms for the factors/outcomes looked at, as well as in the use of their sub-sections’ titles (see below).

Thank you we agree. We have used the general term well-being throughout the document.

Second, row 70-73 report that they also look at barriers and facilitators of the physical activity, but they never describe in details what can be/are those, and which ones they look in their results (Table 3). I also recommend to distinguish between barriers (panel A) and facilitators (panel B) in Table 3.

A description of the barriers and facilitators which are measured can be found in the methods section starting at line 100.

We have clarified the barriers vs. facilitators in table 3 and provided separate subtitles.

Third, there is a complete lack of description of the hypothesis the authors have in mind when they look at the data. What we should expect and why (should more/less active people increase/decrease physical activity after Covid-19)? The authors cite different studies, but a more structured discussion on expected hypothesis might help the reader. If there is an hypothesis worthy to test with descriptive data, please describe it in the introduction. If not, then clarify the paper is a description of the data with ex-ante no expectation on what we should expect.

Overall, I would like to see a revised version of the introduction:

  • To describe exactly what the objectives are

Objectives have been updated as stated above. We appreciate this suggestion.

  • To explain concisely which variables the authors are exploring: what do the authors use for well-being (only mental well-being?), etc.

We have updated the terminology throughout the document to only state well-being in a description of both anxiety and general mental health using the mental health continuum. Additionally, in the objectives we have added well-being (anxiety, general mental health) to provide context as to the variables being assessed.

  • Hypothesis testing based on the literature cited

The following hypotheses have been added to the end of the literature review based on the feedback anf changes to the objectives

 “We hypothesize that COVID-19 would negatively impact physical activity participation overall and that this would be associated with barriers to physical activity. Additionally, we expect this to have a negative impact on Canadian well-being especially amongst those who reduced their physical activity levels. Lastly, we expect that those participants who spend more time being physically active in the outdoors would have greater well-being.”

  1. Organization of the article

Sample

I would like to see many more details on the sample selection and sample in the Sample sub-section. I would move row 144-156  under Sample. It is OK to leave Table 1 as a result since all the paper is descriptive, but at least how many people are in the sample should be described in the sample sub-section.

Thank you. The following statement has been moved up. We felt it was best to leave the remainder of the sample descriptions in the results section so that readers had previously learned about the methods prior to reading.

“While 1366 participants began the online survey, 268 responses were incomplete and therefore removed from the dataset. A resultant participant sample of 1098 Canadian adults were included in this study.”

In addition, the statement “…recruitment through snowball sampling using social media and regular media communications including stories in national and local media” is too general: What was the procedure in practice? How many people were selected, how many reached? How many were excluded because of the criteria? What was the targeted sample? Which social media exactly (internet?)? How do you use stories in the media? …

The following statement has been added to help inform the reader regarding how snowball sampling took place.

“Specifically, the use of personal twitter, facebook and linked in accounts of the authors, as well as stories from the University of the Fraser Valley, the local newspaper, a national information resource center, the outdoor recreation council of BC and a national news network were all involved in spreading the word regarding the study and referring participants to the study link.”

Anyone who was a Canadian and over the age of 19 could complete the survey and therefore it is impossible to know how many were excluded as they would not have completed the survey if they did not meet this criterion. We aimed to target 1000 Canadians in order to have an adequately robust sample but also be able to close the study before restrictions were lessened and the potential impacts of the restrictions altered.

Related to this, could you explain what does “268 incomplete responses” mean? Do you have 0 refusals? How much incompleteness is there? Did participants receive incentives?

There were no incentives involved in this study and therefore participation was entirely voluntary. We do not know if there were refusals as only participants who wanted to participate in the study would have started the study. In relation to this the 268 incomplete responses were individuals who did not complete past the first half of the questionnaire and therefore were removed from the dataset.

Sub-sections

The titles of the sub-sections 3 could better define what the authors look at, following their new objectives above. For example, Titles 3.3 (you have two titles 3.3, so adjust to 3.4) could be changed to reflect “correlations”.

The following sub-section headings have been revised to reflect the objectives.

3.1 Changes to Physical Activity Engagement and Barriers and Facilitators to Physical Activity

3.2 Well-Being and Anxiety Scores based on Changes to Physical Activity 

3.3. Well-Being Scores based on Social Support and Levels of Physical Activity

3.4. Differences in Outdoor Physical Activity and Nature Exposure based on Classifications of Anxiety and Well-Being

Results

The most interesting part of “3.3. Outdoor Physical Activity Behaviour and Well-Being” are rows 235-250: this is what describe “correlations”. Why is Table 5 a descriptive table rather than showing those correlations? I would move Table 5 and rows 226-234 in the first part of the results where sample and changes are described, and dedicate the second part of the paper to correlations. Show the correlations results in an additional table.

Thank you for this comment. We have relocated the descriptive information to section 3.1. Additionally, we have clarified that the objective was to examine differences in outdoor physical activity and nature exposure based on classifications of anxiety and well-being, which does not include any correlation analysis.

  1. Interpretation

What is really lacking is also more details on how you asked the question to define active/inactive people. Is it based on what they were doing before Covid-19? Was it: in a normal week? In the same week last year? etc. The reference time period is fundamental here to understand how the two groups compare.

In addition is 150 minutes weeekly a normal measure used to define active people? 150 minutes seem pretty high and it seems you are really defining “sporty” people rather than just active. So, I would explain better your choice around this definition, to make sure the reader know how to interpret results.

Thank you. You are right this is an important point. This has been further addressed in the methods section for physical activity and the sections underlined were added:

The measure of physical activity was assessed for the week prior and therefore would have been based on physical activity during COVID-19 restrictions. Participants reported their current physical activity levels using the Godin Leisure Questionnaire [12]. In order to assess whether participants were physically active, amounts of reported vigorous and moderate physical activity participation in the Godin questionnaire were used to categorize participants as active or inactive. The cut off for active is > 150 minutes of moderate-vigorous physical activity per week while the related cut off for inactive would be < 149.9 minutes of moderate-vigorous physical activity per week based on standard physical activity guidelines for health-related benefits [13].

Minor comments:

  1. Be consistent: maybe always report “recreation facilities, city parks and playgrounds” not one or the others (row 41, 31-32, 9-10)

Thank you this has been updated

  1. Move reference [3] after the social ecological model, not at the end of the sentence

Thank you this has been updated

  1. April and early May 2020 during the strictest public health restrictions in Canada (row 83) – Please define number of days and when/which restrictions were imposed at the same time

The majority of Canadian provinces implemented similar degrees of restriction beginning March 17 and the survey was closed on May 5 prior to any provincial restrictions being lifted.

The following statement has been added to the manuscript “(nationwide restrictions were in place for 50 days).”

  1. How long was the questionnaire? It seems you asked a lot of questions, maybe describe how long was your survey tool. A strength of the study is that you used well-validated measures. I also like the explanation for each section of measures on how the measures have been used/validated.

Thank you. The questionnaire had an average completion time of 15 minutes so while it was lengthy it didn’t take participants long to complete.

  1. Typo: pos hoc tests: post-hoc? Row 92

Thank you this has been changed.

  1. Typo: row 156 3.1. Physical Activity Engagement and Barriers and Faciltators to Physical Activity

Thank you this has been changed

  1. Typo: 3.2. Mental Well-Being and Phyical Activity

Thank you this has been changed

  1. Typo: 3.3. Phyical Activity, Social Support and Social Well-Being

Thank you this has been changed

  1.  Row 220 repetition …with others to a moderate amount with others scored higher on the mental health continuum

Thank you. The “with others” has been removed.

  1. Funding: This research received no external funding. How is this possible if the authors collected data?

It is not uncommon for smaller institutional research programs to conduct their studies without the use of funding. In the case of this study the investigators completed all of the work on their own time and utilized an existent survey monkey account to collect the data. All recruitment was done through snowball sampling as described with no monetary cost and there were no incentives for participant engagement.

Reviewer 2 Report

Overall, this is a well-written, timely, and important study which informs facilitators and barriers to physical activity during COVID. I have a number of mostly minor comments/recommendations:

Line 10: It is mentioned provincial parks were closed. Were there also restrictions on national parks?

Lines 11-12: Rather than stating the statistical tests that were used in the abstract, it would be better to state what data were collected and what methods were used before presenting the results in the abstract.

It is assumed that increased physical activity levels in people who were normally inactive alleviated anxiety and mental stress. Instead, is it possible that anxiety and mental stress prevented people from getting more active (i.e. if one was stressed about the pandemic, they might be less likely to go outside their house to exercise)? In this case, rather than targeting this population to increase physical activity as a way to reduce stress/anxiety, could you also target this population to reduce stress/anxiety to increase their physical activity levels? Perhaps this can be mentioned at the end of the manuscript in your limitations section.

At the end of the introduction, please include a hypothesis statement or series of hypothesis statements.

Lines 85 to 87: “Demographic characteristics included age, sex, marital status, occupational status (changes due to COVID-19) and childcare obligations (due to COVID-19) were additionally captured.” Please re-write the last part of this sentence or divide into two sentences (or perhaps simply change “included” to “including”).

Line 131: Change “t-test’s” (i.e. possessive) to “t-tests” (i.e. plural).

Line 137: Change “pos hoc” to “post-hoc”. Also, please indicate what type of post-hoc test was used.

Line 139: Please change “ANOVA’s” to “ANOVAs”. Although it is common practice to use an apostrophe + s to indicate plural, this is grammatically incorrect and should be used to indicate possession of something.

In a footnote to Table 1, please clarify what some of the p-values mean. I assume these are differences between the sexes, but in some categories it is unclear where the differences are. For example, for education and employment status, there are multiple sub-categories. Do the significant p-values here refer to all the subcategories or just specific sub-categories? The same question applies to Table 2 – e.g. there are multiple subcategories for outcomes such as “most common type of physical activity”, but only one p-value listed here. Which sub-category does the p-value correspond to?

Lines 168-170: “Regardless of physical activity level there were significant differences between those who became more active, stayed the same or were less active since COVID-19 restrictions and a variety of physical activity barriers and facilitators” – there seems to be something missing from the last part of this sentence or the sentence simply needs re-writing.

Lines 170-171: “To explore between group differences, a series of Tukey’s post hoc tests were conducted.” Please clarify whether this was done after a significant ANOVA.

Lines 219-222: “Specifically, for both the active and inactive populations, those who engaged in physical activity with others to a moderate amount with others scored higher…” – this sentence needs some re-wording (perhaps add the word “compared” before “to”).

For tables 3 and 4, you have p-values presented for each category of “Inactive” and “Active” participants. These p-values indicate the differences within three groups of participants (i.e. “more active”, “same”, “less active”. Please add symbols to the tables to indicate which of these three groups is different (I assume this would be from your post-hoc tests).

Table 5: Again, you have p-values, but it is unclear which comparisons these p-values belong to. For example, the first p-value for “physical activity location” has three locations described. Which of these does the p-value correspond to?

Line 271: Change “becomes” to “became”

Author Response

Reviewer 2 Responses

Comments and Suggestions for Authors

Overall, this is a well-written, timely, and important study which informs facilitators and barriers to physical activity during COVID. I have a number of mostly minor comments/recommendations:

Line 10: It is mentioned provincial parks were closed. Were there also restrictions on national parks?

 Yes there were. This has been updated to read “city, provincial and national parks”

Lines 11-12: Rather than stating the statistical tests that were used in the abstract, it would be better to state what data were collected and what methods were used before presenting the results in the abstract.

This line in the abstract has been updated to read:

“An online survey was utilized to measure participant physical activity behavior, nature exposure, well-being and anxiety levels.”

It is assumed that increased physical activity levels in people who were normally inactive alleviated anxiety and mental stress. Instead, is it possible that anxiety and mental stress prevented people from getting more active (i.e. if one was stressed about the pandemic, they might be less likely to go outside their house to exercise)? In this case, rather than targeting this population to increase physical activity as a way to reduce stress/anxiety, could you also target this population to reduce stress/anxiety to increase their physical activity levels? Perhaps this can be mentioned at the end of the manuscript in your limitations section.

We agree that it is a chicken or the egg question. We have added the following statement in the conclusions section:

“This suggests that health promoting measures directed towards inactive individuals may be essential to improving well-being or alternatively improving well-being of Canadians in order to increase physical activity levels.”

At the end of the introduction, please include a hypothesis statement or series of hypothesis statements.

Thank you. The following has been added

 “We hypothesize that COVID-19 would negatively impact physical activity participation overall and that this would be associated with barriers to physical activity. Additionally, we expect this to have a negative impact on Canadian well-being especially amongst those who reduced their physical activity levels. Lastly, we expect that those participants who spend more time being physically active in the outdoors would have greater well-being.”

Lines 85 to 87: “Demographic characteristics included age, sex, marital status, occupational status (changes due to COVID-19) and childcare obligations (due to COVID-19) were additionally captured.” Please re-write the last part of this sentence or divide into two sentences (or perhaps simply change “included” to “including”).

Rewritten as the following:

“Demographic characteristics included age, sex and, marital status. Additionally, occupational status (changes due to COVID-19) and childcare obligations (due to COVID-19) were captured.”

Line 131: Change “t-test’s” (i.e. possessive) to “t-tests” (i.e. plural).

This has been changed

Line 137: Change “pos hoc” to “post-hoc”. Also, please indicate what type of post-hoc test was used.

 Thank you this has been changed to read “Tukey post-hoc tests”

Line 139: Please change “ANOVA’s” to “ANOVAs”. Although it is common practice to use an apostrophe + s to indicate plural, this is grammatically incorrect and should be used to indicate possession of something.

 This has been changed

In a footnote to Table 1, please clarify what some of the p-values mean. I assume these are differences between the sexes, but in some categories it is unclear where the differences are. For example, for education and employment status, there are multiple sub-categories. Do the significant p-values here refer to all the subcategories or just specific sub-categories? The same question applies to Table 2 – e.g. there are multiple subcategories for outcomes such as “most common type of physical activity”, but only one p-value listed here. Which sub-category does the p-value correspond to?

Thank you for this comment. The following footnote has been added to Tables 1 and 2:

 “The p-values represent chi-square tests of independence indicating associations between sex and categorical variables.”

Lines 168-170: “Regardless of physical activity level there were significant differences between those who became more active, stayed the same or were less active since COVID-19 restrictions and a variety of physical activity barriers and facilitators” – there seems to be something missing from the last part of this sentence or the sentence simply needs re-writing.

Thank you for this clarification. This has been rewritten as the following:

“Regardless of physical activity level there were significant differences between those who became more active, stayed the same or were less active since COVID-19 restrictions. Within these differences there were differing roles of physical activity barriers and facilitators (see Table 3 for summary).”

Lines 170-171: “To explore between group differences, a series of Tukey’s post hoc tests were conducted.” Please clarify whether this was done after a significant ANOVA.

This has been updated to the following:

 To explore between group differences, a series of Tukey post hoc tests were conducted after a significant between group difference was found.

Lines 219-222: “Specifically, for both the active and inactive populations, those who engaged in physical activity with others to a moderate amount with others scored higher…” – this sentence needs some re-wording (perhaps add the word “compared” before “to”).

Thank you this has been updated. There was a repeat of the “with others”

For tables 3 and 4, you have p-values presented for each category of “Inactive” and “Active” participants. These p-values indicate the differences within three groups of participants (i.e. “more active”, “same”, “less active”. Please add symbols to the tables to indicate which of these three groups is different (I assume this would be from your post-hoc tests).

Subscripts have been added to these tables and the following note included below the table:

“Means in a row sharing subscripts are significantly different from each other”

Table 5: Again, you have p-values, but it is unclear which comparisons these p-values belong to. For example, the first p-value for “physical activity location” has three locations described. Which of these does the p-value correspond to?

 Thank you for this comment. The following footnote has been added to Tables 5 (which has become Table 3):

“The p-values represent chi-square tests of independence indicating associations between sex and categorical variables.”

Line 271: Change “becomes” to “became”

This has been changed

Reviewer 3 Report

The study covers a hot topic that in this period is worth being addressed. The study is linear, well-written, well-presented, and easy to read.  I would like to congratulate the Authors for their work, that of course is worth being shared with the scientific community.

I have only a point that I suggest to address: did the Authors consider whether in their sample there were also athletes? I believe it is possible that a portion of their sample could be composed by athletes. They certainly were part of the active group, but they were quite different as compared to people engaging in physical activity for well-being purposes, rather than for competing.

Author Response

Reviewer 3 Responses

Comments and Suggestions for Authors

The study covers a hot topic that in this period is worth being addressed. The study is linear, well-written, well-presented, and easy to read.  I would like to congratulate the Authors for their work, that of course is worth being shared with the scientific community.

I have only a point that I suggest to address: did the Authors consider whether in their sample there were also athletes? I believe it is possible that a portion of their sample could be composed by athletes. They certainly were part of the active group, but they were quite different as compared to people engaging in physical activity for well-being purposes, rather than for competing.

Thank you for this comment. We did not purposefully sample athletes in our study but they likely would have been a part of our sample. It is challenging to define someone as an athlete using the assessment of moderate-vigorous activity as an individual could be an avid walker and accumulate a large amount of moderate physical activity but not be an athlete. This is something that would be interesting to look into in another study as likely athletes have been negatively affected by the restrictions put in place and likely the associated well-being impacts.

Round 2

Reviewer 1 Report

Thank you for addressing my comments. The manuscript was improved and it can now be accepted.